# The Impact of Long-Term Clinoptilolite Administration on the Concentration Profile of Metals in Rodent Organisms

**DOI:** 10.3390/biology12020193

**Published:** 2023-01-26

**Authors:** Ivan Dolanc, Lejla Ferhatović Hamzić, Tatjana Orct, Vedran Micek, Iva Šunić, Antonija Jonjić, Jasna Jurasović, Saša Missoni, Miran Čoklo, Sandra Kraljević Pavelić

**Affiliations:** 1Centre for Applied Bioanthropology, Institute for Anthropological Research, 10000 Zagreb, Croatia; 2Institute for Medical Research and Occupational Health, 10000 Zagreb, Croatia; 3Institute for Anthropological Research, 10000 Zagreb, Croatia; 4Faculty of Dental Medicine and Health, University of Osijek, 31000 Osijek, Croatia; 5Faculty of Health Studies, University of Rijeka, Viktora Cara Emina 5, 51000 Rijeka, Croatia

**Keywords:** heavy metals, clinoptilolite, ICP-MS

## Abstract

**Simple Summary:**

Human activities such as heavy industry and transport have significantly increased the levels of many poisonous chemicals in the human environment. Among these chemicals are heavy metals, which pose a great risk to human health because they cannot decompose and cannot be eliminated from the human body by metabolic processes. Volcanic stone called clinoptilolite is inert and extremely porous, and can capture heavy metals into its meshed structure. Therefore, it can be used as a food supplement for detox purposes. Several clinical studies have already indicated its detoxifying, antioxidant and anti-inflammatory effects. In our experiment, we fed healthy rats with clinoptilolite, which was milled to fine dust to improve performance, for three months. Two forms were used: tribomechanically activated zeolite and Panaceo-Micro-Activated zeolite. Upon completion of the administration period, we observed the impact of clinoptilolite dust on the metallic composition in different rat tissues and bloodstream. Our results showed that this activated natural stone had indeed released metallic toxicants from the rat organs to the bloodstream, which indicates a detoxification process.

**Abstract:**

Heavy metals are dangerous systemic toxicants that can induce multiple organ damage, primarily by inducing oxidative stress and mitochondrial damage. Clinoptilolite is a highly porous natural mineral with a magnificent capacity to eliminate metals from living organisms, mainly by ion-exchange and adsorption, thus providing detoxifying, antioxidant and anti-inflammatory medicinal effects. The in vivo efficiency and safety of the oral administration of clinoptilolite in its activated forms, tribomechanically activated zeolite (TMAZ) and Panaceo-Micro-Activated (PMA) zeolite, as well as the impact on the metallic biodistribution, was examined in healthy female rats. Concentration profiles of Al, As, Cd, Co, Pb, Ni and Sr were measured in rat blood, serum, femur, liver, kidney, small and large intestine, and brain using inductively coupled plasma mass spectrometry (ICP-MS) after a 12-week administration period. Our results point to a beneficial effect of clinoptilolite materials on the concentration profile of metals in female rats supplemented with the corresponding natural clinoptilolite materials, TMAZ and PMA zeolite. The observed decrease of measured toxicants in the kidney, femur, and small and large intestine after three months of oral intake occurred concomitantly with their most likely transient release into the bloodstream (serum) indicative of a detoxification process.

## 1. Introduction

Metals are ubiquitous and naturally occurring in the Earth’s crust and atmosphere, but their concentrations have recently risen to extreme levels in the environment. This is to a great extent a byproduct of human activities such as agriculture, heavy industry, transport, cosmetics, natural medicaments and the beauty industry [1,2,3]. Of particular concern is the highly increased abundancy of heavy metals, characterized by high atomic weight and a density of at least 5 g/cm^3^. These metals adversely affect the environment and living organisms. Generally, they are considered toxic, depending on the chemical species, dose and route of exposure, as well as on the age, gender, pharmacogenetics and nutritional status of exposed persons. Of the metals, arsenic (As), cadmium (Cd), chromium (Cr) and lead (Pb) are highly dangerous [4]. Considered as systemic toxicants, they are capable of inducing multiple organ damage even at low concentrations, primarily by the induction of oxidative stress and mitochondrial damage [4,5].

Metal toxicity can be acute or chronic, depending on the absorbed dose, and the route and duration of exposure. Even though prolonged exposure to low quantities of heavy metals and even aluminum (Al), classified as a light metal, in the short-term are not considered to be generally noxious (e.g., via cosmetics), various toxic effects including allergic contact dermatitis or systemic toxicity have been documented [6]. A few metals can be removed from living organisms by metabolic elimination (e.g., Al), but the majority of metals accumulate in the body, imposing a long-term health risk. Mechanistically, they bind to proteins by displacing original metals from their natural bidding sites and cause the malfunctioning of proteins, cells and tissues. The oxidative deterioration of biomolecules is primarily due to the binding of metals to DNA and nucleic proteins [7]. Consequential health perturbations may include osteoporosis, carcinogenesis, neurodegeneration, oxidative stress, endocrine disruption, DNA damage and immune system deterioration, organ failure and disturbances in reproduction [2,8,9,10].

Although the individual health hazards of Pb, Cd and mercury (Hg) exposure are extensively studied, more attention should be given to the examination of their synergistic effect, since combined exposure to these toxicants in environmentally relevant concentrations causes damage to multiple organs as well as impairment of the neurobehavioral functions of rats [11,12].

Several metals are considered essential and nutritionally valuable, since they play an important role in human physiology. Other metals, often referred to as heavy metals, such as As, Cd and Pb, are common food contaminants, with unknown biological functions and considerable negative impact on human health [1,13,14]. Some biological consequences documented for heavy metals include inflammation, oxidative stress, endocrine disruption and intestinal disorders correlated with gut microflora perturbations, and compositional and metabolic profile changes. Probiotic strains and their enzymes may help in the alleviation of these effects [15]. Some trace minerals (also belonging to metals) exhibit beneficial effects in vivo and may ameliorate metallic toxicity. One such element is selenium (Se), which sequesters As and Cd and incorporates them into inert complexes. Moreover, Se induces the erythroid 2-related factor 2 (Nrf2) pathway. This highly biologically active protein is a potent regulator of metallic toxicity. It triggers protective oxidative stress mechanisms through the release of antioxidants [16,17].

Importantly, the basic harmfulness to health of Al mainly arises from its pro-oxidant activity and consequential oxidative stress that leads to degradation of cellular proteins and lipids. Chronic excessive exposure to Al may provoke oxidative stress, tissue inflammation, genotoxicity, carcinogenesis, teratogenesis, tissue necrosis, endocrine disruption, diabetes onset, obesity, inhibition of cartilage and bone formation, hypertension, ischemic stroke and thrombosis [18]. Circulating Al is usually bound to transferrin, which seems to potentiate Al entrance in the central nervous system in a manner similar to iron [19]. Accordingly, increased Al levels in the brain are detrimental, and high interconnections of Al deposition in the brain are in correlation with neurotoxic effects, pathogenesis of Alzheimer disease [20,21] and autism spectrum disorders [22]. Thus, Al detoxification might be considered as a possible therapeutic approach in these conditions, as already seen in patients with Alzheimer’s disease [23].

Arsenic toxicity emerges from its extensive binding affinity and inhibition of selenoenzymes, which are known scavengers of reactive oxygen species [24]. This causes multiple changes in cell behavior through alterations in signaling pathways and epigenetic modifications, as well as direct oxidative damage to proteins and lipids [25]. Acute As poisoning causes abdominal pain and gastrointestinal problems [26]. Prolonged As exposure may cause skin lesions, development of neoplasms, peripheral neuropathy, cardiovascular disease and neurodevelopmental problems [24]. However, in addition to its toxic effect, As has certain beneficial medical properties. Specifically, As trioxide greatly improves the survival of patients suffering from acute promyelocytic leukemia [27]. Such applications fall into a specific and controlled medical therapy field.

Cadmium possesses no known biological function in humans, but has a very low rate of excretion and tends to accumulate in organs (e.g., brain, liver, kidney and testes), causing their impairment. It reduces the reproduction of both sexes even at low doses [28]. Its strongest toxic effects are seen in kidneys. Cd tends to accumulate in this organ as a result of its preferential uptake by receptor-mediated endocytosis in the proximal tubules. This triggers proteinuria, which can progress to renal Fanconi syndrome and eventually renal failure [29,30]. To exert the toxic effect, Cd must enter the intracellular space [31], where it triggers oxidative, apoptotic, spermal and steroidogenic toxicity [32]. Moreover, this metal exerts immunomodulatory function, and can trigger the release of chemokines, regulate gene expression and attenuate inflammation [33]. Although extensive research is being conducted, no efficient Cd antidotes are available for clinical application so far [34].

Cobalt (Co) is biologically relevant for the formation of vitamin B-12, but excessive exposure to this metal correlates with some pathological conditions such as the development of reduced thyroid activity, interstitial pulmonary fibrosis and cardiomyopathy [35]. Humans are usually exposed to Co via four distinct routes: occupational, environmental, dietary and medical. Of these, oral intake and internal exposure through hip implants are the most common medical hazards for the development of Co systemic toxicity. The release of free ionic Co^2+^ upon prolonged excessive exposure to this element seems to underlie cobalt-induced chronic poisoning [36]. In microorganisms, Co is highly receptive to iron sulfur proteins, and, perturbing iron homeostasis, it incites oxidative stress [37].

Lead is not biodegradable, and when absorbed in organisms, it accumulates in blood, bones, liver, kidney, brain and skin. Negative effects on the reproductive, hepatic, endocrine, immune and gastrointestinal systems have been reported in humans [38]. Pb possesses strong affinity for sulfhydryl groups and electron donor groups. Thus, it tightly binds to various proteins, causing multiple harms to the affected individuals. It also competes with nutrients such as calcium (Ca) or zinc (Zn) in various important mechanisms which are usually mediated by their ions. Pb easily crosses cell membranes, exerts redox reactions and induces the formations of reactive oxygen species [39]. Thus, the toxic effects of Pb are quite complex and affect virtually all organs [40]. In adults, it can increase blood pressure, slow nerve conduction or induce fatigue, mood swings, drowsiness, impaired concentration, infertility, decreased libido, headaches, constipation or encephalopathy, or even cause death in severe cases [41]. Chronic exposure in early childhood causes adverse neurodevelopmental consequences even at low levels [42], and multi-organ damage or even death at high doses [43,44].

The hazard posed by nickel (Ni) to human health originates primarily from the production of free radicals, generation of the superoxide anion and reactive oxygen species. The consequential oxidative stress triggers genotoxicity, carcinogenicity and immunotoxicity [45]. Additional hazard mechanisms of Ni toxicity include competition with essential metals in metalloproteins or binding to enzymes and inhibition of their activity [46]. Ni is shown to induce various pathophysiological processes in humans, including those involved in the development of allergy, cardiovascular and kidney diseases, lung fibrosis, and pulmonary, nasal and sinus cancer. This metal is associated with epigenetic effects, namely DNA hypermethylation and histone ubiquitination, which are the key mechanisms associated with tumor initiation, cancer progression and metastasis [47].

Strontium (Sr) is a trace element in the human body but has never been proved to be essential. Interestingly, the toxic effects of Sr overdose have not been reported in humans [48]. This mineral naturally accumulates in mineralized tissues by surface exchange or ionic substitution mechanisms. However, the exaggerated accumulation of Sr in bone is shown to be associated with development of osteomalacia in dialysis patients [49]. This element exerts remarkable osteogenic potential since it promotes bone formation and inhibits bone resorption. Thus, it has been incorporated in various orthopedic devices to enhance bone regeneration, especially in osteoporotic patients, where it improves bone mineral density [50,51,52,53].

Special consideration has to be given to the potential of zeolites’ usage for the removal of metal pollutants. These naturally occurring, highly-porous minerals, also known as molecular sieves, show enormous capacity to eliminate metals from polluted environments, mainly by mechanisms known as ion-exchange and adsorption [54,55]. This property has been proved for zeolite clinoptilolite in both animals and humans [56]. The crystalline structure of these aluminosilicates has cavities filled with cations and polar molecules, such as water. These chemical entities are released from the structure and simultaneously substituted by chemical species from the environment, depending on their zeolite binding affinity [57]. Moreover, zeolite clinoptilolite is highly inert and safe for human utilization [58]. Both the preclinical and clinical literature imply the great medical potential of this material in the treatment of various diseases [58,59,60]. The volcanic mineral zeolite clinoptilolite is the most widely used natural zeolite in medicine. Prior to usage, it is usually activated by grinding into fine powder. In this process, its active surface and activity increase substantially [61]. Certain types of clinoptilolites have already been certified for clinical usage [56]. One such certified material, therefore determined as safe for oral human applications, is PMA (Panaceo-Micro-Activated) zeolite, with its detoxifying, antioxidant and anti-inflammatory properties [62].

In this work, we examined the in vivo efficiency and safety of the long-term oral administration of zeolite clinoptilolite materials in natural and activated forms. We intentionally used an excessively high dose of this zeolite (8 g/kg) to examine its effect on high metallic biodistribution in vivo. Our focus was on the measurement of the concentration profiles of Al and several contaminants (As, Cd, Co, Pb, Ni and Sr) in healthy rats fed with zeolite clinoptilolite materials. Metal concentrations were measured in the brain, gastrointestinal and excretory systems, bones and circulation systems of the animals.

## 2. Materials and Methods

### 2.1. Zeolites

Clinoptilolite materials were obtained from Panaceo International Gmbh (Villach/Gödersdorf, Austria). The same natural clinoptilolite tuff was used in the form of tribomechanically activated zeolite (TMAZ), milled using a standard tribomechanical micronization, and double tribomechanically activated zeolite clinoptilolite, processed using a patented tribomechanical micronization production method (PMA zeolite). The latter patented technology for double activation uses multiple high-speed particle collision through whirlwinds that are generated by seven circular rows of blades fixed on two counter-rotating discs, wherein the counter-rotating discs are arranged such that the particles have to pass all seven circular rows of blades by centrifugal force. The velocity of the blades is 145 m/s. The above-described activation process is executed two consecutive times (‘double-activation’). It is aimed at increasing the particle temperature and particle collisions, giving rise to new surface properties described in more detail in Kraljevic Pavelic et al. [61]. The detailed physical–chemical properties and preparation of PMA zeolite are also presented in Kraljevic Pavelic et al. [61]. Ludox AS-40 colloidal silica was purchased from Sigma-Aldrich chemicals Co. (St. Louis, MO, USA).

### 2.2. Animals

The experiment was conducted at the Institute for Medical Research and Occupational Health in Zagreb. Female HsdBrlHan: Wistar rats (200–220 g) were bred and maintained under pathogen-free conditions in a steady-state microenvironment and fed with a standard GLP certified laboratory chow Mucedola 4RF21 (Mucedola, Settimo Milanese, Italy) and tap water ad libitum. Additionally, the rats were exposed to a 12/12 h light schedule. All the experimental procedures were performed following the Guide for the Care and Use of Laboratory Animals: Eighth Edition (National Academies Press, 2010).

### 2.3. Drugs and Treatment Schedule

Clinoptilolite and Ludox suspensions were prepared fresh daily prior to oral administration (gavage). Two different forms of clinoptilolite were daily administered by gavage as water suspension at a dose of 8 g/kg of body weight. Colloidal silica was also administered daily at a dose of 2 g/kg of body weight. The study was carried out for a period of three months (12 weeks). The rats were randomly assigned to four different experimental groups:Group 1 (n = 10): I (received drinking water);Group 2 (n = 10): TMAZ;Group 3 (n = 10): PMA zeolite;Group 4 (n = 10): colloidal silica (Ludox AS-40).

### 2.4. Tissue Sampling

After the three-month administration protocol, the rats were anesthetized with intraperitoneal injections of Narketan (80 mg/kg of body mass) and Xylapan (12 mg/kg of body mass) (Chassot AG, Bern, Switzerland) and sacrificed by exsanguination. Blood, serum, femur, liver, kidney, small and large intestine, and brain were harvested for further analysis by ICP-MS.

### 2.5. Inductively Coupled Plasma Mass Spectrometry (ICP-MS)

The presence of metals (Al, As, Cd, Co, Ni, Pb and Sr) was analyzed in the blood, serum and harvested tissues at the Analytical Toxicology and Mineral Metabolism Unit of the Institute for Medical Research and Occupational Health in Zagreb, using inductively coupled plasma mass spectrometry (ICP-MS) on the Agilent 7500cx Series (Agilent Technologies, Waldbronn, Germany). A mixture of nitric acid (65%, purity, m.p., Merck, Germany) and deionized water in equal volumes (2 mL) was added to 0.5 g of tissue samples and mixed vigorously. The obtained suspension was microwaved under high pressure in the UltraCLAVE IV high-pressure microwave device (Milestone, S.r.l. Italy) following the program for biological samples. The calibration curves of standards prepared in 1% HNO3 (*v*/*v*) from the mono-element standard solution (PlasmaCAL, CP Science, Canada) were used for the quantification of the elements. The sensitivity of the ICP-MS instrument was adjusted by a solution containing 1 μg/L of lithium (Li), magnesium (Mg), cobalt (Co), yttrium (Y), cerium (Ce), thallium (Tl) and selenium (Se). The response of detectors at masses 7 (Li), 89 (Y) and 205 (T1) was monitored, at the same time the ratios of doubly charged ions (^140^Ce^2+^/^140^Ce^+^) and the oxides of the elements (^140^Ce^16^O^+^/^140^Ce^+^) were compared with the single-charged ions. Optimal conditions are achieved when satisfactory sensitivity (by varying the voltage on the lens, the depth of the sample input and the gas flow rate of the carrier) is obtained, with minimal formation of doubly charged ions (<2.2%) and oxide ions (<1.4%). For the accuracy and precision of the ICP-MS method, the following commercially available reference blood materials were used: Recipe ClinChek^®^ Whole blood lyophilized Level I, II, III and SeronormTM Trace Element Whole Blood Level I, II for blood; Recipe ClinChek^®^ Serum Control lyophilized Level I, II, Plasma Control lyophilized Level I, II and SeronormTM Trace Elements Serum Level I, II for serum/plasma; and BRM-IAEA-H5 Animal Bone for bone. All the reference materials were reconstituted by the addition of ultrapure water according to the manufacturers’ protocols, and were used to confirm the accuracy and precision of the method. The bone material reference sample was prepared in the same way as the biological samples of the rat organs. The physiological levels of the measured metals and minerals in the rats are given in Appendix A.

### 2.6. Statistical Analysis

Statistical analysis was performed using GraphPad Prism 9 (GraphPad Software, San Diego, CA, USA). Statistical differences between groups were examined using a *t*-test (*t*-test: two samples assuming equal variances). The data were presented as arithmetic mean ± standard deviation.

## 3. Results

The long-term administration of TMAZ and PMA zeolite, as well as Ludox silica, induced distinct, statistically relevant changes in the concentrations of the examined metals, depending on the analyzed tissue. Overall, the majority of the significant changes in metal levels were observed in the serum, while no major changes were observed in the brain. The kidney and liver as major detoxification organs showed increased concentrations of Pb with a concomitant decrease of Pb levels in the large intestine upon PMA zeolite administration. In the kidney, small and large intestine, and femoral bone, the majority of the changes pointed to decreased levels of the analyzed metals. Specifically, Al levels were altered in the kidney, serum and femur (Figure 1). TMAZ reduced this metal level in the kidney and serum. Both TMAZ and PMA zeolite, as well as Ludox, showed similar effects on the change in Al concentration profiles in the tested rats. These zeolites increased Al levels in the serum while concomitantly decreasing its levels in the femur. Ludox decreased Al levels in the kidney as well. This may be indicative of Al release from bone deposits.

As levels changed in the small intestine and serum (Figure 2). TMAZ reduced the level of this metal in the small intestine, while both PMA zeolite and Ludox increased its levels in serum. Overall, Cd levels were reduced in the presented experiment (Figure 3). PMA zeolite reduced its levels in the small intestine and serum.

Of the analyzed metals, Co was the only metal which did not show any concentration changes in the experiment (Figure 4). Ni was, however, the only metal whose levels were increased in the femur, as observed in all three experimental groups (Figure 5). PMA zeolite reduced Ni levels in the kidney and blood, which was accompanied by an increase in levels in the serum. Ludox induced same changes, except for the kidney, where it did not cause significant changes in Ni levels.

Pb levels changed upon treatments in the kidney, liver, large intestine, femur and blood (Figure 6). TMAZ reduced its levels in the blood, while PMA zeolite increased its levels in the kidney and liver with a concomitant decrease in the large intestine. Ludox induced a decrease of Pb levels only in the femur.

Sr levels changed in the large intestine, serum and blood (Figure 7). Its levels were increased in the serum in all three experimental groups. TMAZ increased its levels in the blood, while Ludox reduced its levels in the large intestine and the blood.

## 4. Discussion

Zeolites are increasingly considered as an efficient approach in the fight against the toxicological burden of metallic pollution. Owing to their unique physicochemical characteristics, and most importantly, their ion-exchange and adsorption properties, zeolites possess magnificently powerful capabilities for decontamination and the reduction of metallic burden, either in industrial applications or in animals and humans. The first applications with zeolite clinoptilolite for decontamination were performed more than 30 years ago [63,64]. Since then, zeolite clinoptilolite has shown a high affinity with metallic pollutants in many studies [56,63,64,65], and has been extensively used for the elimination of Al, Pb, Cd, Sr, Co, Ni, Mg, Cr and As from contaminated areas. Besides industrial applications, zeolite clinoptilolite is increasingly used in veterinary and human medicine, as it has been proved to be safe, inert and resilient to metabolism [59]. Accordingly, diverse in vivo effects were documented as well, including antioxidant, hemostatic, anti-diarrhetic, immunomodulatory and detoxification properties [66]. More importantly, in several recent clinical trial studies on humans with a natural clinoptilolite material (PMA zeolite), it has been shown that preloaded metals do not enter the blood stream from the intestine, but that the observed fluctuations of metal levels in the blood are a consequence of the activation of detoxification processes from various body compartments [67]. The main mechanisms underlying these biological effects include incorporation of metallic ions within the zeolite clinoptilolite crystal lattice, but additional biological effects should be considered as well. For example, the zeolite clinoptilolite structure has been found to neutralize free radicals by trapping them within the complex structure, which leads to their chemical inactivation [66]. This is highly relevant, since metals exert their toxic effects in humans primarily through the massive induction of free radicals and triggering of oxidative stress processes. As clinoptilolite materials are zeolites of natural origin, they are mined as structures pre-loaded with a variety of elements, including metals. The pre-requisite for their in vivo oral application is, accordingly, strict material control [56].

In the presented paper, we obtained data that underlines previously observed PMA zeolite properties in Al elimination from the body [61]. Indeed, the Al levels in the femur of PMA zeolite-supplemented animals were significantly decreased, probably due to the bone remodeling process resulting in a transient increase in the serum. The same effect was observed in a pilot study in patients with irritable bowel syndrome [68]. The bone remodeling process has already been observed in clinoptilolite treatment—specifically with PMA zeolite treatment in vivo [69] —and is at least partially attributable to soluble silicon species release from the zeolite clinoptilolite material into the blood. This is in line with our results, which are comparable in animals treated only with colloidal silica. Furthermore, on the basis of data in the literature, activated bone remodeling process and high affinity of the clinoptilolite material towards Pb, we may hypothesize that Pb-detoxification process from the bone compartment has been triggered in rats supplemented with PMA zeolite. This process is usually very slow and may last for years [70,71]. Indeed, increased Pb levels were observed in the kidneys and liver of the TMAZ-supplemented animals as well, with the same trend but without significance. This is expected, as these organs represent the main detoxification routes for Pb. In the large intestine, Pb levels were statistically decreased in PMA zeolite-treated animals. The similarity of effects for TMAZ and PMA zeolite, but a stronger, statistically relevant change observed for PMA zeolite, may be attributable to the smaller particle size and larger active surface area in the PMA zeolite material [57]. Importantly, rats have a coprophage behavior [72], and serum Pb in our experiment may come both from body deposition departments such as bones but partially also from re-ingested feces.

Furthermore, the PMA zeolite material statically decreased the levels of Cd in the serum and in the small intestine, as well as the levels of Ni in the kidneys. Interestingly, PMA zeolite increased levels of Ni in the femur, but this increase did not exceed physiological levels observed in female rats [73]. In the context of previously observed bone remodeling process occurring in clinoptilolite treated animals and humans [69], this may be correlated with an anabolic effect of zeolite on bone tissue, as is the case for Zn and Co as well [74]. Moreover, levels in the femur were reduced only in the TMAZ-supplemented animals. The velocity of effects on the observed metal elimination from deposits in the body seems indeed to be correlated with the material’s physical–chemical properties. It was shown previously that in humans, for example, PMA zeolite significantly decreased As upon longer PMA zeolite intake, specifically a 12-week supplementation in the clinical trial Morbus Crohn study [67].

Finally, Sr levels increased in the serum of animals supplemented with TMAZ, PMA zeolite and Ludox. This might point to a connection or even increased uptake of Sr along with Si from the intestine owing to soluble silica forms released from the zeolites and present in Ludox in all the applied supplementations. Recently, Xing et al., for example, reported that Si and Sr ions released from bioceramic hydrogels synergistically stimulated cell proliferation and stimulate osteogenic differentiation [75]. However, further investigation of this phenomena possibility in vivo should be conducted in more detail.

## 5. Conclusions

Our results point to a beneficial effect of clinoptilolite materials on the concentration profile of metals in female rats supplemented with the corresponding natural clinoptilolite materials, TMAZ and PMA zeolite. The observed decrease of measured toxicants in the kidney, femur, and small and large intestine after three months of oral intake occurred concomitantly with their most likely transient release into the bloodstream (serum), indicative of a detoxification process. The similarity of effects for TMAZ and PMA zeolite, but with a stronger, statistically relevant change observed for PMA zeolite for Pb mobilization from the body compartments, may be attributable to the particle size and active surface increase in the PMA zeolite material. This phenomenon has been also observed in humans within a controlled clinical trial and merits further study, as it may be relevant for the improvement of bone quality.

## Figures and Tables

**Figure 1 biology-12-00193-f001:**
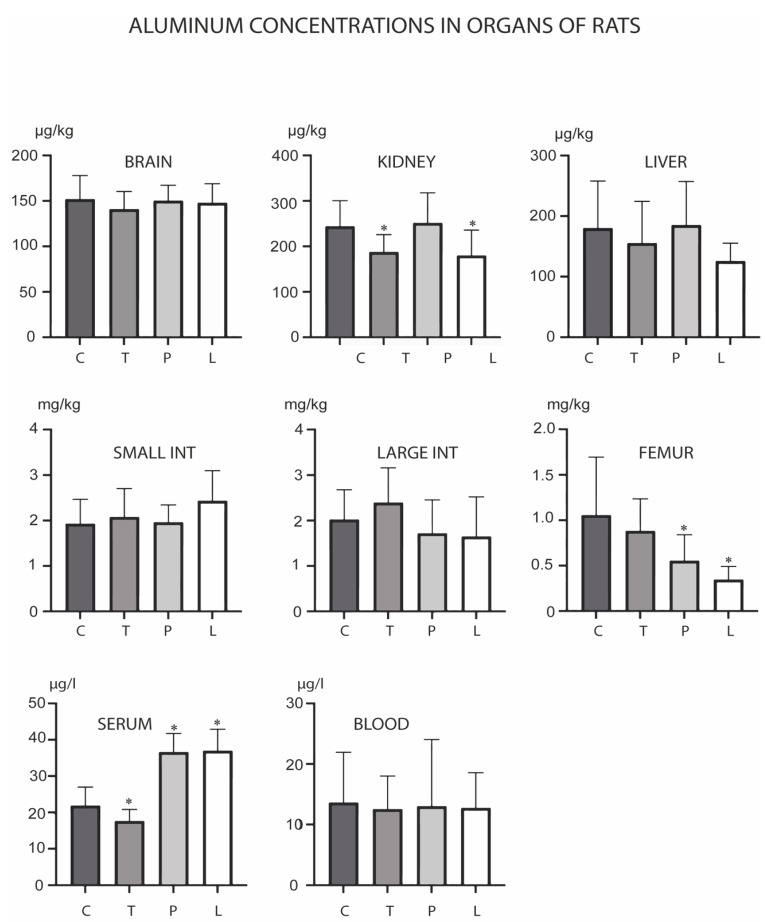
Concentrations of aluminum in the brain, kidney, liver, small intestine, large intestine, serum, blood and femur after the three-month oral administration of silicates. C denotes the control group; T denotes the experimental group receiving tribomechanically activated zeolite clinoptilolite (TMAZ); P denotes the experimental group receiving the double tribomechanically activated zeolite clinoptilolite (PMA zeolite); and L denotes the experimental group receiving Ludox silica. Asterisk (*) denotes significant differences between the experimental groups, with control at *p* < 0.05.

**Figure 2 biology-12-00193-f002:**
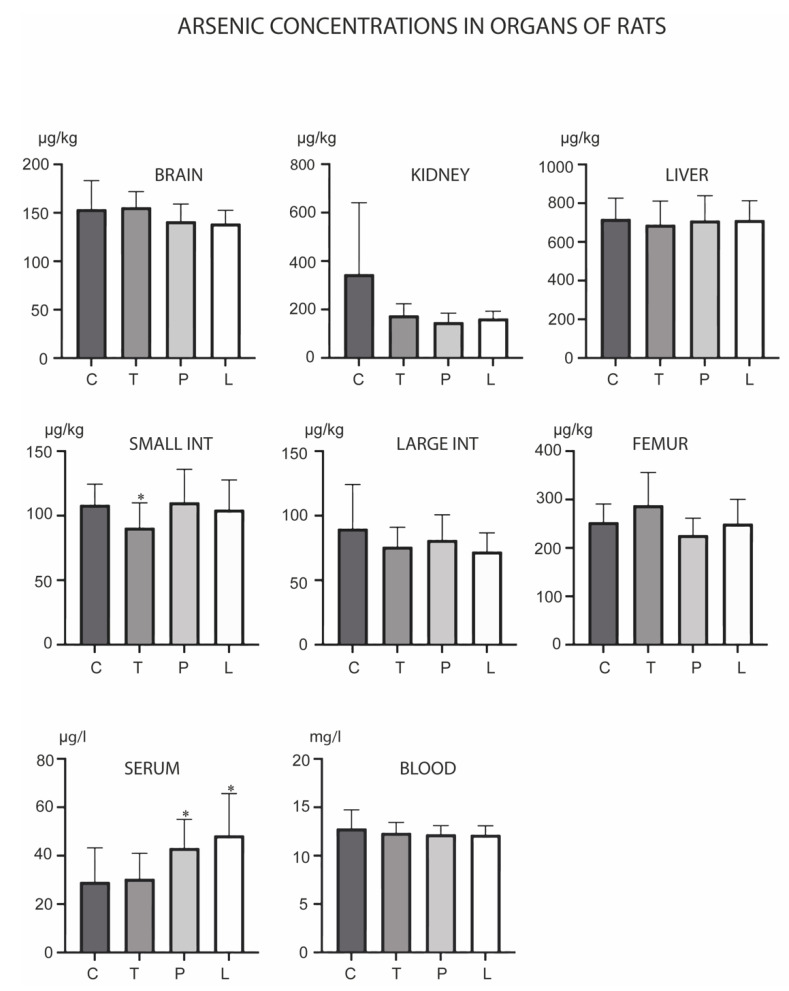
Concentrations of arsenic in the brain, kidney, liver, small intestine, large intestine, serum, blood and femur after the three-month oral administration of silicates. C denotes the control group; T denotes the experimental group receiving TMAZ; P denotes the experimental group receiving PMA; and L denotes the experimental group receiving Ludox silica. Asterisk (*) denotes significant differences between the experimental groups, with control at *p* < 0.05.

**Figure 3 biology-12-00193-f003:**
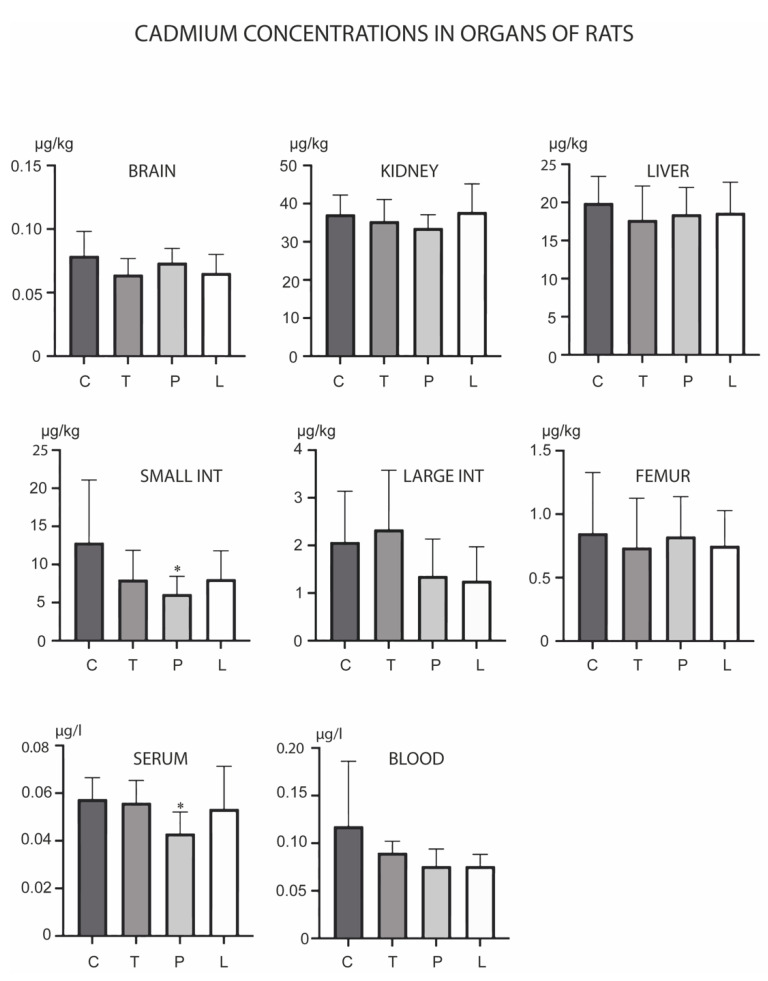
Concentrations of cadmium in the brain, kidney, liver, small intestine, large intestine, serum, blood and femur after the three-month oral administration of silicates. C denotes the control group; T denotes the experimental group receiving TMAZ; P denotes the experimental group receiving PMA; and L denotes the experimental group receiving Ludox silica. Asterisk (*) denotes significant differences between the experimental groups, with control at *p* < 0.05.

**Figure 4 biology-12-00193-f004:**
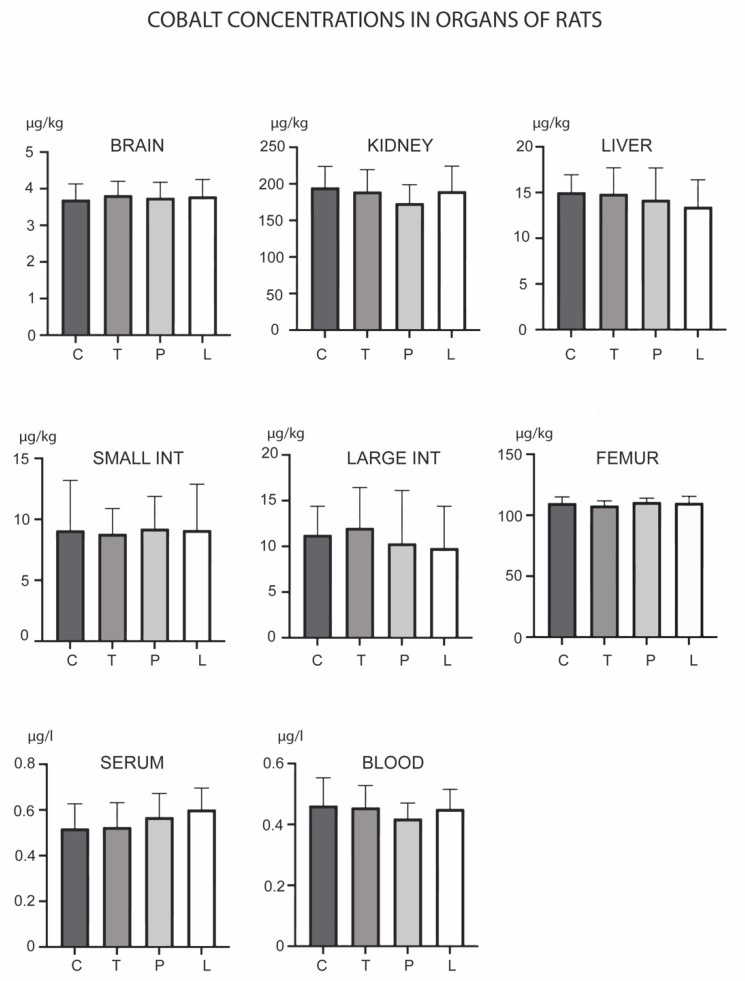
Concentrations of cobalt in the brain, kidney, liver, small intestine, large intestine, serum, blood and femur after the three-month oral administration of silicates. C denotes the control group; T denotes the experimental group receiving TMAZ; P denotes the experimental group receiving PMA; and L denotes the experimental group receiving Ludox silica. Asterisk (*) denotes significant differences between the experimental groups, with control at *p* < 0.05.

**Figure 5 biology-12-00193-f005:**
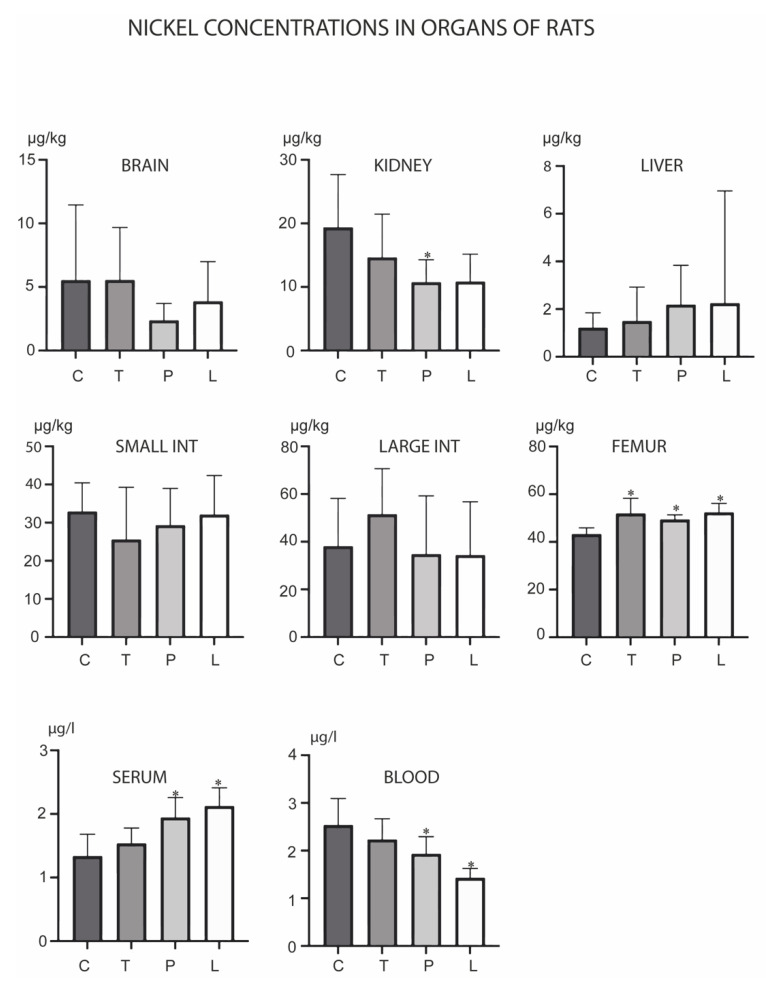
Concentrations of nickel in the brain, kidney, liver, small intestine, large intestine, serum, blood and femur after the three-month oral administration of silicates. C denotes the control group; T denotes the experimental group receiving TMAZ; P denotes the experimental group receiving PMA; and L denotes the experimental group receiving Ludox silica. Asterisk (*) denotes significant differences between the experimental groups, with control at *p* < 0.05.

**Figure 6 biology-12-00193-f006:**
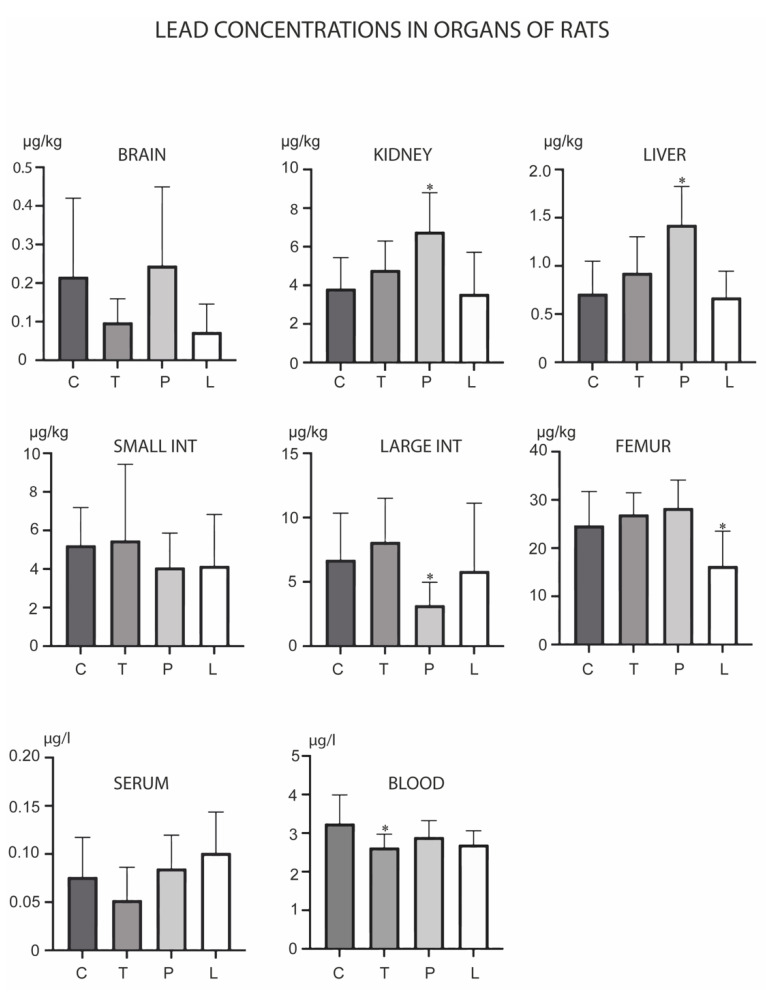
Concentrations of lead in the brain, kidney, liver, small intestine, large intestine, serum, blood and femur after the three-month oral administration of silicates. C denotes the control group; T denotes the experimental group receiving TMAZ; P denotes the experimental group receiving PMA; and L denotes the experimental group receiving Ludox silica. Asterisk (*) denotes significant differences between the experimental groups, with control at *p* < 0.05.

**Figure 7 biology-12-00193-f007:**
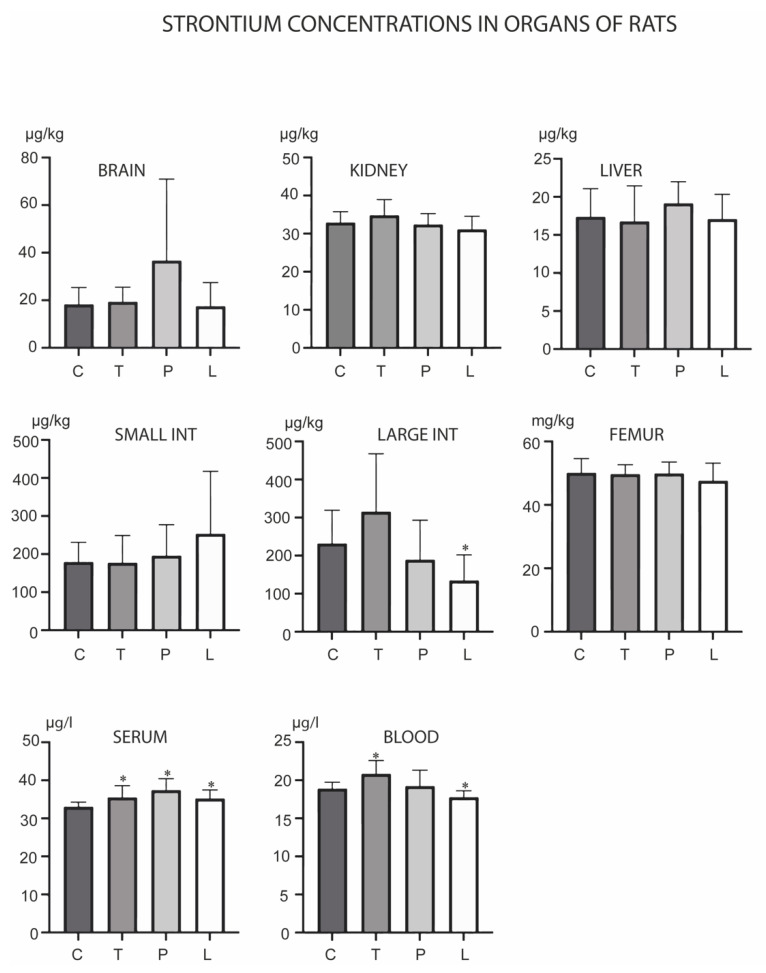
Concentrations of strontium in the brain, kidney, liver, small intestine, large intestine, serum, blood and femur after the three-month oral administration of silicates. C denotes the control group; T denotes the experimental group receiving TMAZ; P denotes the experimental group receiving PMA; and L denotes the experimental group receiving Ludox silica. Asterisk (*) denotes significant differences between the experimental groups, with control at *p* < 0.05.

## Data Availability

Not applicable.

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
