# Peer review of "The Impact of Long-Term Clinoptilolite Administration on the Concentration Profile of Metals in Rodent Organisms"

_biology, 2023, doi:10.3390/biology12020193_

Round 1

Reviewer 1 Report

The article is devoted to the interesting issue of metals redistribution in the organs and tissues during prolonged natural mineral sorbents exposure. Concentration сhanges of Al, As, Cd, Co, Pb, Ni, Sr in the brain, kidneys, liver, intestines, bones, tissues and in blood were studied. The research method - Inductively coupled plasma mass spectrometry (ICP-MS) is accurate enough to obtain reliable results in the chosen experiment design. Methods are described clearly and in sufficient detail.

However, there are some remarks:

1.    The figures reflect the results, but are of low and varying resolution, asymmetrically arranged and looks sloppy. Illustration design needs to be revised.

2.  For a complete understanding of the situation, it is desirable to provide information about the presence of tested metals in the structure of natural zeolites used in the study. Since these minerals may contain various metals, which the authors themselves emphasize in lines 371-373.

Author Response

Reviewer 1

Comments and Suggestions for Authors

The article is devoted to the interesting issue of metals redistribution in the organs and tissues during prolonged natural mineral sorbents exposure. Concentration сhanges of Al, As, Cd, Co, Pb, Ni, Sr in the brain, kidneys, liver, intestines, bones, tissues and in blood were studied. The research method - Inductively coupled plasma mass spectrometry (ICP-MS) is accurate enough to obtain reliable results in the chosen experiment design. Methods are described clearly and in sufficient detail.

Thank you for your observations.

However, there are some remarks:

  1. The figures reflect the results, but are of low and varying resolution, asymmetrically arranged and looks sloppy. Illustration design needs to be revised

We have now revised all figures according to the comments and saved them with an improved resolution.

  1. For a complete understanding of the situation, it is desirable to provide information about the presence of tested metals in the structure of natural zeolites used in the study. Since these minerals may contain various metals, which the authors themselves emphasize in lines 371-373.

The materials are all derived from the same raw material subject to regular testing and chemical analysis. These data from official datasheets is now included in the supplementary material.

Reviewer 2 Report

This is an interesting in vivo study of the effects of administered micronised clinoptilolite in rats. Silica and drinking water were given to control groups. Some of these authors are long term researchers of the effects of zeolites in animals, with several reviews and toxicology studies published.

The Introduction is very well written and cited. 

The present study is interesting in a preliminary sense, but it asks more questions than it answers. The Methods lacks details. What exactly is the difference between TMAZ and PMA? Is it simply double-timed micronisation? Whatever it is, are there sizing profiles for the nanoparticles? Two large or too small will affect results by the passage or non-passage of particles into cells. 

Some of the silica results mimicked the clinoptilolite results, but there was a lack of consistency in the results for organs and fluids that cannot simply be explained by the TMAZ and PMA. Knowing their sizing profiles would help. 

Also, I find the trial design flawed. Only single groups (n = 10) of female rats were used, but we have no prior evidence of measurements of the range of heavy metals before the study began. There could be quite wide fluctuations in levels of metals among rats in each group and between groups.

Taking a single measurement point at 12 weeks is also open to challenge. Why 12 weeks? Why not have additional groups of rats at (n = 6) and test the metals levels at 4 weeks, 8 weeks, and 12 weeks?

As far as it goes, this study suggests some effects but the design is too narrow to make significant conclusions. Patchy results are evident so that some seemingly good data is clouded by non-responsive effects.

It is a lightweight study in its present form and would benefit greatly by being expanded in scope and methodology.

Author Response

This is an interesting in vivo study of the effects of administered micronised clinoptilolite in rats. Silica and drinking water were given to control groups. Some of these authors are long term researchers of the effects of zeolites in animals, with several reviews and toxicology studies published.

The Introduction is very well written and cited. 

The present study is interesting in a preliminary sense, but it asks more questions than it answers. The Methods lacks details. What exactly is the difference between TMAZ and PMA? Is it simply double-timed micronisation? Whatever it is, are there sizing profiles for the nanoparticles? Two large or too small will affect results by the passage or non-passage of particles into cells.

Thank you for your comments. Indeed, the presented study is part of our group larger efforts in research on in vivo effects of tested zeolite materials. One of our first published papers indeed, provides more data on the physical-chemical properties of the herein tested zeolite materials (Kraljević Pavelić et al. (2017) Novel, oxygenated clinoptilolite material efficiently removes aluminium from aluminium chloride-intoxicated rats in vivo. Novel, oxygenated clinoptilolite material efficiently removes aluminium from aluminium chloride-intoxicated rats in vivo // Microporous and mesoporous materials, 249 (2017), 146-156 doi:10.1016/j.micromeso.2017.04.062 ). In addition to this relevant reference provided in the material and methods section, we have now included more explanation on the micronization methods in the material and methods section as well.

Some of the silica results mimicked the clinoptilolite results, but there was a lack of consistency in the results for organs and fluids that cannot simply be explained by the TMAZ and PMA. Knowing their sizing profiles would help.

We have now included data on the particle size in the Supplementary material.

Also, I find the trial design flawed. Only single groups (n = 10) of female rats were used, but we have no prior evidence of measurements of the range of heavy metals before the study began. There could be quite wide fluctuations in levels of metals among rats in each group and between groups.

We have measured the physiological metal concentrations in the rats used for our in vivo studies. These values are now given in the supplementary material.

Taking a single measurement point at 12 weeks is also open to challenge. Why 12 weeks? Why not have additional groups of rats at (n = 6) and test the metals levels at 4 weeks, 8 weeks, and 12 weeks?

Thank you for this comment. We opted for the long term evaluation of the effects of mineral homeostasis as this issue raises major interest among clinicians that use clinoptilolite materials in the clinical setup.

As far as it goes, this study suggests some effects but the design is too narrow to make significant conclusions. Patchy results are evident so that some seemingly good data is clouded by non-responsive effects.

It is a lightweight study in its present form and would benefit greatly by being expanded in scope and methodology.

Thank you for this comment. Indeed, we study the presented materials’ in vitro and in vivo effects for more than 10 years and obtained data is being published continuously and timely upon completion of each research projects. The whole picture is accordingly, we agree on that, more easily understandable in a larger context. This is why we opted to published all our experimental data to shed more light on this interesting inorganic material. Some of our other papers on zeolites include:

  • Kraljević Pavelić S*, Micek V, Filošević A, Gumbarević D, Žurga P, Bulog A, Orct T, Yamamoto Y, Preočanin T, Plavec J, Peter R, Petravić M, Vikić-Topić D, Pavelić K (2017): Novel, oxygenated clinoptilolite material efficiently removes aluminium from aluminium chloride-intoxicated rats in vivo. Micropor Mesopor Mat 249(1); 146-156
  • Kraljević Pavelić S*, Simović Medica S, Gumbarević D, Filošević A, Pržulj N, Pavelić K. (2018) Critical Review on Zeolite Clinoptilolite Safety and Medical Applications in vivo. Frontiers in pharmacology 9:1350, https://doi.org/10.3389/fphar.2018.01350
  • Kraljević Pavelić S*, Micek V, Bobinac D, Bazdulj E, Gianoncelli A, Krpan D, Žuvić M, Eisenwagen S, Stambrook PJ, Pavelić K (2020) Treatment of osteoporosis with a modified zeolite shows beneficial effects in an osteoporotic rat model and a human clinical trial, Experimental biology and medicine, 246(5): 529–537.
  • Kraljević Pavelić S*, Krpan D, Žuvić M, Eisenwagen S, Pavelić K (2022) Clinical parameters in osteoporosis patients supplemented with PMA-zeolite at the end of 5-year double-blinded clinical trial, Front Med, 9: article number 870962, https://doi.org/10.3389/fmed.2022.870962
  • Kraljević Pavelić S*, Saftić Martinović L, Simović Medica J, Žuvić M, Perdija Ž, Krpan D, Eisenwagen S, Orct T (2022) Clinical evaluation of a defined Zeolite-Clinoptilolite supplementation effect on the selected blood parameters of patients, Front Med (Lausanne). 2022; 9: article number 851782

Reviewer 3 Report

Dear Authors,

In my opinion the theme of the manuscript is innovate and very interesting for the readers of the journal. 

The authors administrated during 12 weeks and measured concentration profiles of Al, As, Cd, Co, Pb, Ni and Sr in rat blood, serum, femur, liver, kidney, small and large intestine and brain with inductively coupled plasma mass spectrometry after a period of administration.

It was observed that clinoptilolite is highly porous natural mineral with magnificent capacity to eliminate metals from living organisms, mainly by ion-exchange and adsorption, and thus providing the detoxifying, antioxidant, and anti-inflammatory medicinal effects.

The in vivo efficiency and safety of the oral administration of clinoptilolite in its activated forms: Tribomechanically Activated Zeolite (TMAZ) and Panaceo Micro Activated (PMA) zeolite, as well as their impact on the metallic biodistribution was examined in healthy female rats.

Results highlighted to a beneficial effect of clinoptilolite materials on the concentration profile of metals in female rats supplemented with the corresponding natural clinoptilolite materials zeolite TMAZ and zeolite-PMA.

The observed decrease of measured toxicants in the kidney, femur, small and large intestine after three months of oral intake, occurs concomitantly with their, most likely transient, release in the bloodstream (serum) indicative for a detoxification process. The similarity of effects for zeolite TMAZ and zeolite- PMA, but a stronger, statistically relevant change observed for zeolite-PMA for Pb mobilization from the body compartments, may be attributable to the particle size and active surface increase in the zeolite-PMA material.

The authors highlighted to a beneficial effect of clinoptilolite materials on the concentration profile of metals in female rats supplemented with the corresponding natural clinoptilolite materials zeolite TMAZ and zeolite-PMA.

The manuscript under revision is well structured, the language is correct and clear and the title and abstract clearly describe the content of the manuscript. The authors should remove references in the Conclusion section.

In my opinion the manuscript is almost ready to be published. Please take my comments in consideration and add the proposed reference, in attached revised file.

Congratulations!

Best regards

Author Response

Reviewer 3.

Dear Authors,

In my opinion the theme of the manuscript is innovate and very interesting for the readers of the journal. 

The authors administrated during 12 weeks and measured concentration profiles of Al, As, Cd, Co, Pb, Ni and Sr in rat blood, serum, femur, liver, kidney, small and large intestine and brain with inductively coupled plasma mass spectrometry after a period of administration.

It was observed that clinoptilolite is highly porous natural mineral with magnificent capacity to eliminate metals from living organisms, mainly by ion-exchange and adsorption, and thus providing the detoxifying, antioxidant, and anti-inflammatory medicinal effects.

The in vivo efficiency and safety of the oral administration of clinoptilolite in its activated forms: Tribomechanically Activated Zeolite (TMAZ) and Panaceo Micro Activated (PMA) zeolite, as well as their impact on the metallic biodistribution was examined in healthy female rats.

Results highlighted to a beneficial effect of clinoptilolite materials on the concentration profile of metals in female rats supplemented with the corresponding natural clinoptilolite materials zeolite TMAZ and zeolite-PMA.

The observed decrease of measured toxicants in the kidney, femur, small and large intestine after three months of oral intake, occurs concomitantly with their, most likely transient, release in the bloodstream (serum) indicative for a detoxification process. The similarity of effects for zeolite TMAZ and zeolite- PMA, but a stronger, statistically relevant change observed for zeolite-PMA for Pb mobilization from the body compartments, may be attributable to the particle size and active surface increase in the zeolite-PMA material.

The authors highlighted to a beneficial effect of clinoptilolite materials on the concentration profile of metals in female rats supplemented with the corresponding natural clinoptilolite materials zeolite TMAZ and zeolite-PMA.

The manuscript under revision is well structured, the language is correct and clear and the title and abstract clearly describe the content of the manuscript. The authors should remove references in the Conclusion section.

In my opinion the manuscript is almost ready to be published. Please take my comments in consideration and add the proposed reference, in attached revised file.

Thank you for your kind comments and suggested literature that we think increases the value of our manuscript and has been added accordingly. We have removed now the reference from the conclusion section.

Round 2

Reviewer 2 Report

Revised version is satisfactory.